

# Deep learning-based methodology for vulnerability detection in smart contracts

Zhibo Wang[1], Liu Guoming[2], Hongzhen Xu[1], Shengyu You[3], Han Ma[2] and Hongling Wang[1]

[1] College of Information Engineering, East China University of Technology, Nanchang, Jiangxi, China
[2] Department of Computer Science and Technology, East China University of Technology, Nanchang, Jiangxi, China
[3] School of Software, East China University of Technology, Nanchang, Jiangxi, China

## ABSTRACT

Smart contracts play an essential role in the handling and management of digital assets, where vulnerabilities can lead to severe security issues and financial losses. Current detection techniques are largely limited to identifying single vulnerabilities and lack comprehensive identification capabilities for multiple vulnerabilities that may coexist in smart contracts. To address this challenge, we propose a novel multi-label vulnerability detection model that integrates extractive summarization methods with deep learning, referred to as Ext-ttg. The model begins by preprocessing the data using an extractive summarization approach, followed by the deployment of a custom-built deep learning model to detect vulnerabilities in smart contracts. Experimental results demonstrate that our method achieves commendable performance across various metrics, establishing the effectiveness of the proposed approach in the multi-vulnerability detection tasks within smart contracts.

## INTRODUCTION

The concept of "smart contracts" was first introduced by computer scientist Nick Szabo in 1994 (*Szabo, 1997*). In his seminal article, Szabo depicted contracts grounded in computer protocols, or contract clauses that could be automatically executed through computer code. However, for an extended period, the practical application and progress of smart contracts encountered a bottleneck due to the absence of a trustworthy execution environment. Fortunately, the emergence of blockchain technology has radically transformed this situation. With its decentralized nature, immutability, and transparency, blockchain technology has provided a reliable execution environment for smart contracts, facilitating their effective application in technology. Presently, smart contracts have been successfully implemented in various domains, including financial services, healthcare, the Internet of Things (IoT), and education, emerging as one of the cornerstone technologies of blockchain (*Alfuhaid et al., 2023*).

Corresponding authors
Zhibo Wang, zhbwang@ecut.edu.cn, 23076608@qq.com
Liu Guoming, 2339545214@qq.com, 2022110203@ecut.edu.cn

Numerous blockchain platforms, such as Ethereum, EOS, and Fabric, have integrated smart contracts as an indispensable component. Despite the widespread adoption of smart contracts in blockchain, the ensuing security risks have become increasingly prominent. Initially, smart contracts are written in languages like Solidity, where code-writing is inherently prone to vulnerabilities that can lead to non-compliance with the contract's intended behavior. Moreover, given that smart contracts govern significant assets such as digital currencies, malicious actors exploit these vulnerabilities to orchestrate attacks and expropriate digital currencies, resulting in substantial economic losses. To date, there have been several cases of major economic loss. For instance, the 2016 DAO incident (*Mehar et al., 2019*), where attackers exploited a reentrancy vulnerability, inserting attack code into the contract execution flow, and repeatedly calling the contract's functions to siphon nearly 60 million. On April 22, 2018, hackers capitalized on an integer overflow vulnerability to attack the BEC's Token contract (*Etherscan, 2018*), transferring a massive number of Tokens to exchanges, which nearly plummeted BEC's price to zero. In 2019, the synthetic asset platform Lendf.me suffered a reentrancy attack, resulting in a theft of 25 million. Once deployed on a blockchain, a smart contract's code and state are perpetually stored across various blocks. If a smart contract is compromised, the attack is permanently recorded on the blockchain and is irreversible. This immutable nature makes it challenging for smart contracts to revert to a pre-attack state. To safeguard blockchain security, it is crucial to perform vulnerability detection on smart contracts before deployment.

The application of deep learning in program analysis has increased notably in recent years, offering high automation and speed while overcoming the limitations of rule-based vulnerability detection methods. Deep learning approaches automatically learn and extract latent features from raw data and offer excellent scalability, which has led to significant achievements when combined with smart contract vulnerability detection. However, existing deep learning-based methods for smart contract vulnerability detection still have some drawbacks: (1) most can only distinguish between vulnerable and non-vulnerable contracts (a binary classification problem) or detect a single type of vulnerability, ignoring that many contracts contain multiple vulnerabilities. (2) Feeding overly long sequences of smart contract opcodes directly into models not only incurs significant time costs but also can result in decreased accuracy.

To address the aforementioned issues, this article introduces an enhanced multi-label vulnerability detection method, Ext-ttg, which integrates an extractive summarization technique with a custom-designed neural network (transformer + Bi-GRU). The method begins by transforming smart contract source code into opcode sequences, as delineated in the Ethereum yellow article. An extractive summarization approach is then employed to distill key sequences from the opcodes, mitigating the impact of excessive opcode lengths on feature extraction for the model. Subsequently, the opcode sequences are vectorized using word2vec, which facilitates the extraction of semantic information through the bespoke ttg model. Ultimately, a sigmoid classifier is utilized to perform multi-label classification of vulnerabilities in smart contracts. The primary contributions of this article are as follows: (1) An extractive summarization method is utilized to capture key sequences from opcode sequences, enabling the representation of opcode sequences of varying lengths in

a standardized form. This assists the model in better learning the critical features within the opcode sequences, thus enhancing the model's generalization capabilities. (2) Given the complexity of the opcode sequences, the multi-head attention mechanism within transformers is employed to focus on the positional and semantic information of opcodes in the sequences. This aids in capturing both global and local semantic information. To further enhance the capture of global features, a bi-directional gated recurrent unit (Bi-GRU) network is integrated subsequent to the transformer layer. (3) Comparisons with other deep learning networks are conducted, affirming the efficacy of the proposed model.

## RELATED WORK

The detection of vulnerabilities in smart contracts can be primarily categorized into several methodological approaches: symbolic execution, fuzz testing, formal verification, intermediate representation, and deep learning.

Existing smart contract vulnerability detection methods fall into three broad categories: manual inspection, automated detection, and AI-based detection. Manual inspection relies on the experience and skill level of the reviewer, which is subjective and inefficient, failing to meet the demands of smart contract verification. Consequently, the development of automated detection tools is crucial for research in smart contract vulnerability detection. To date, automated detection methods mainly consist of four types: symbolic execution, fuzz testing, formal verification, and intermediate representation.

Symbolic execution aims to identify potential vulnerable input conditions by simulating program execution paths. It abstracts input values symbolically rather than using specific predetermined inputs, enabling smart contract analysis. Representative tools include Oyente (*Luu et al., 2016*), Securify (*Tsankov et al., 2018*), Mythril (*Ethereum, 2017*), and Orisi (*Torres, Schütte & State, 2018*). Fuzz testing uncovers potential security issues by monitoring for abnormal results in execution states using copious amounts of random or semi-random test data. Key tools are ContractFuzzer (*Jiang, Liu & Chan, 2018*) and ILF (*He et al., 2019*). Formal verification employs mathematical logic to ensure that code meets certain properties under conditions described by specifications. It involves modeling smart contracts in formal language, then using mathematical reasoning to identify security vulnerabilities. Notable tools in this category are Zeus (*Kalra et al., 2018*) and VaaS (*Beosin, 2019*). Intermediate representation involves translating smart contract source code or bytecode into an intermediary format that conveys its semantics, followed by compiler analysis to identify security issues, with tools like Slither (*Feist, Grieco & Groce, 2019*), Vandal (*Brent et al., 2018*), and Smartcheck (*Tikhomirov et al., 2018*) being prominent examples.However, current automated detection methods still face challenges, including low accuracy and poor scalability, and they struggle to address unknown vulnerabilities.

As artificial intelligence progresses, numerous researchers have made significant achievements in the field of smart contract vulnerability detection using machine learning and deep learning techniques. *Wang et al. (2021)* constructed an automatic detection model for smart contracts named ContractWard, which extracts bigram features from

smart contract opcodes using the n-gram algorithm, followed by the employment of five machine learning algorithms to build a vulnerability detection model. *Tann et al. (2018)* transformed smart contract source code into opcodes, treated them as textual sequences, and utilized LSTM to extract features between opcode sequences, achieving results with greater precision than the vulnerability detector Maian. *Rossini, Zichichi & Ferretti (2023)* processed smart contract bytecode into RGB images and employed convolutional neural networks (CNN) for image recognition, attaining commendable results on a dataset with five categories of vulnerabilities. Furthermore, *Zhuang et al. (2021)* converted smart contract source code into contract graphs, standardized these graphs through a node elimination process, and then implemented a Degreeless Graph Convolutional Network (DR-GCN) and a Temporal Message Propagation Network (TMP) to detect three different types of vulnerabilities. Subsequently, *Liu et al. (2021)* refined their approach by integrating expert rules with graph neural networks, further enhancing detection accuracy. *Guo, Lu & Li (2024)* captured global and local information from smart contract source code by employing transformers and CNNs, respectively, and ultimately used a Deep Residual Shrinkage Network (DRSN) to detect three types of vulnerabilities, achieving commendable results. *Zhang et al. (2023)* extracted global features from the token sequences of smart contract source code and captured deep structural semantics from the abstract syntax trees of smart contracts, ultimately classifying the combined features using TextCNN. *Sun et al. (2023)* proposed a framework called ASSBert, which utilizes the active and semi-supervised bidirectional encoder representations of Transformer, classifying smart contract vulnerabilities using a small amount of labeled code data and a large volume of unlabeled code data. However, most existing models are limited in the variety of vulnerability categories they can detect, with some focusing exclusively on a single type of vulnerability, unable to identify multiple categories within a smart contract. Therefore, in this article, we use a extractive summarization method to truncate opcodes while retaining significant segments, and develop a custom model based on transformer bi-directional gated recurrent unit to extract semantic features from smart contracts and perform multi-label classification of vulnerability types.

# METHODS

To effectively identify smart contract vulnerabilities, this experiment utilizes a self-built model that combines extractive summarization methods to detect vulnerabilities. The methodology of this research is structured into two primary components: (1) Enhanced Opcode Pruning Strategy Using Extraction-Based Summarization: In this stage, an extractive summarization approach is applied to truncate the opcode sequences. This step is crucial for maintaining consistent lengths across opcodes, which is imperative for the uniformity of feature extraction and subsequent modeling. (2) Multi-label Vulnerability Detection: The core analytical phase employs a composite model that synergizes transformers with bi-GRU. This model framework is designed to extract sequential patterns and semantic features from the opcode sequences. Following the feature extraction, a multi-label classification process is conducted to identify and categorize the

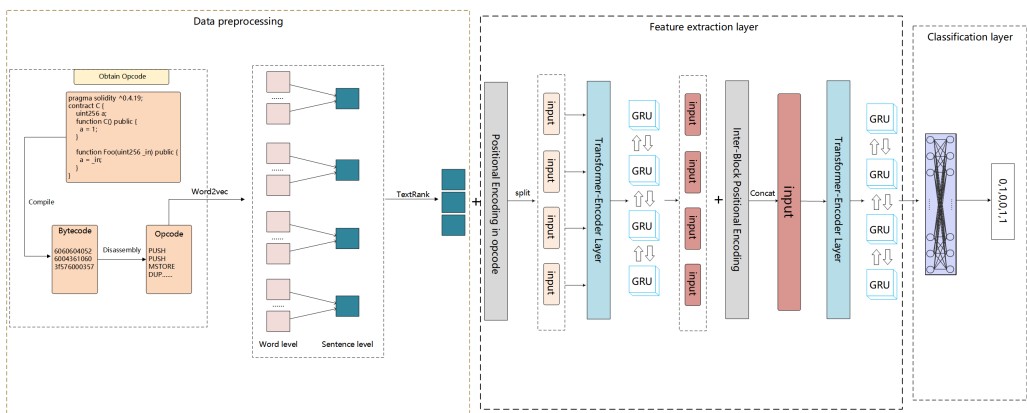

**Figure 1** **The complete structure of the model.** The model employs a stacked Transformer Encoder and Bi-GRU to classify vulnerabilities in opcode sequences trimmed by TextRank.

vulnerabilities present within the smart contracts. The overall model structure is illustrated in Fig. 1.

## Extractive summarization-driven opcode trimming strategy

Smart contract opcode sequences typically extend to a length of approximately eight thousand characters, posing a significant challenge for deep learning networks to effectively extract features from such extensive sequences. In contrast, extractive summarization (*El-Kassas et al., 2021*) methods are techniques that directly extract essential sentences, phrases, or words from the original text, encapsulating the core content into concise summaries. Therefore, this article utilizes Textrank to extract and refine the opcode sequences. TextRank (*Mihalcea & Tarau, 2004*), a graph-based ranking algorithm, evaluates the significance of graph vertices through a holistic, recursive computation that encompasses both localized vertex-specific information and holistic graph-wide intelligence. This article harnesses TextRank to trim opcodes judiciously, maintaining vital opcode sequences with minimal semantic loss.

**Vectorization of opcode sequences:** In the absence of pre-trained models tailored to smart contract opcode texts, we vectorize opcode sequences utilizing the Word2Vec scheme (*Mikolov et al., 2013*). Word2Vec, a model for word embedding, translates words into vector representations, capturing their semantic relationships by considering the words' contextual usage within the corpus. This model features two primary architectures: Continuous Bag of Words (CBOW) and Skip-gram. CBOW predicts target words based on surrounding context, while Skip-gram anticipates the surrounding context from the target words.After experimental analysis, we found that using skop-gram is more effective. So, our experimental analysis favors the Skip-gram model for vectorizing opcode sequences.

**Pruning of opcode sequences:** The initial step in our analytical methodology involves taking a sequence of opcodes, denoted as $B$, and transforming each opcode into a numerical vector *via* the Word2Vec model, resulting in a vectorized sequence $B_v$. The experiment generates opcode vectors with a dimension of 100 using word2vec, yielding a final

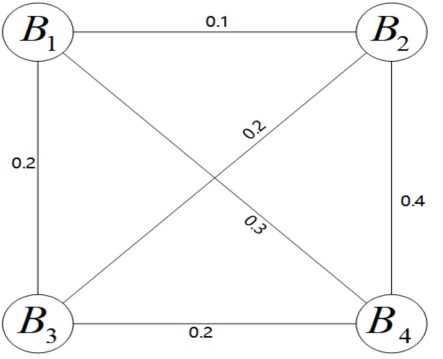

**Figure 2  Undirected graph representation of segmented opcode sequences.** In the graph, a higher edge weight indicates stronger relevance between two opcode sequences, enabling the model to recognize the importance of key sequences.

$B_v$ shape of $[B_{num}, 100]$, where $B_{num}$ indicates the count of opcodes in the sequence. Subsequently, $B_v$ undergoes a segmentation procedure where every group of 128 opcode vectors is averaged to form a segmented vector representation, denoted as $B_{vi}$, such that $B_{vi} = \{b_1, b_2, \ldots, b_{100}\}$. This segmentation effectively reduces the dimensionality of the data, aiding in the subsequent analysis phase. The final step is the computation of a similarity matrix, $Sim_B$, for the segmented vectors $B_{vi}$. This matrix captures the pairwise similarities between segments of the opcode sequence. $Sim_B$ is then converted into a graph structure, $G$, which is input into the TextRank algorithm. TextRank processes $G$ to identify and extract the top 16 segments with the highest scores. These segments are sequentially concatenated to form a coherent, condensed representation of the original opcode sequence. In the graph $G$, each node corresponds to a $B_{vi}$, and the edges represent the similarity scores $Sim_{Bij}$ between segments. This method allows for an efficient and structured analysis of opcode sequences, facilitating the identification of patterns and characteristics pertinent to smart contract vulnerabilities. Figure 2 illustrates the relationships between certain nodes.

$$Sim_{Bij} = \frac{B_{vi} \bullet B_{vj}}{\max(\|B_{vi}\|_2 \bullet \|B_{vj}\|_2)} \text{ where i} \neq \text{j} \tag{1}$$

$$\|B_{vi}\| = \sqrt{\sum_{i=1}^{n} b_i^2} \text{ where n is the vector dimension.} \tag{2}$$

## Multi-label vulnerability detection

This research introduces a proprietary model architecture for feature extraction from vectorized opcode sequences. The process commences with the summation of the vectorized opcode sequence and its correspondingly position-encoded sequence. This step is critical as it imbues the opcode sequence with positional context, which is paramount for capturing the sequence's order-dependent characteristics. Subsequently, the opcode sequence is partitioned and fed in batches into the feature extraction layer. This layer is designed to

distill segment-level information, capturing the nuances of each partitioned section of the opcode sequence. Following this, segment position encoding is integrated before the sequences are concatenated, which is then routed back into the feature extraction layer to derive global context. This approach ensures that both localized and holistic features are considered by the model, enhancing the overall representational capacity. The process ends at the final classification layer, which interprets the extracted features to determine the vulnerability type. This classification constitutes the final step in the model's pipeline, enabling the identification of specific vulnerabilities within the smart contract code based on the learned opcode sequence features.

**Positional embedding:** In sequence data processing, models often handle multiple data points concurrently. To preserve the sequential order information, it is common practice to incorporate positional encoding vectors into word embedding vectors. This technique allows the network to retain the position information of each word while extracting semantic features. Our research utilizes positional encoding that employs sine and cosine functions of different frequencies to encode positional information. Positional encoding serves to provide context about the position of elements within a sequence, enhancing the model's ability to understand order-dependency and sequence relationships. This approach to positional encoding capitalizes on the periodic properties of trigonometric functions to differentiate positional embeddings. For position pos and dimension i, the encoding functions are defined as follows:

$$PE_{(pos,2i)} = \sin(pos/10000^{2i/d}) \tag{3}$$

$$PE_{(pos,2i+1)} = \cos(pos/10000^{2i/d}). \tag{4}$$

In the context of positional embedding, *pos* represents the position of a word—or an opcode in this case—within a sequence, while $d$ denotes the dimensionality of the Positional Embedding *PE*, which is identical to the dimensionality of the opcode sequence. Furthermore, $2i$ and $2i+1$ refer to even and odd dimensions within the embedding, respectively.

**Transformer Encoder:** The Transformer model, introduced by *Vaswani et al. (2017)* in 2017, has revolutionized the field of neural networks with its attention-based mechanism. This architecture distinguishes itself from its predecessors by assigning variable weights to different positions within a sequence, thereby effectively capturing the interdependencies among various positions. Comprising two core components, the encoder and the decoder, the Transformer facilitates a nuanced understanding of sequential data.

The encoder's primary function is to convert the input sequence into a context-aware representation, enriching the model's comprehension of the semantic and structural nuances within the input sequence. It does so through a series of self-attention and feed-forward operations, which allow the model to process all sequence positions simultaneously and weigh them based on their relevance to each other.

In contrast, the decoder is principally tasked with transforming this context-aware representation into a target sequence. However, in the context of this work, our focus is on

the encoder component of the Transformer architecture. We harness the encoder to extract and contextualize information from opcode sequences. By deploying the Transformer Encoder, we can obtain a rich, nuanced understanding of the opcode sequence, which is vital for subsequent tasks such as vulnerability detection or feature extraction in smart contracts. The structure of a single Transformer Encoder unit is depicted in Fig. 3.

Given a vectorized opcode sequence denoted as $E^l = \{e_1^l, e_2^l, \ldots, e_n^l\}$, where $E^l$ signifies the output of the $l-th$ opcode in the sequence. Each Transformer Encoder is composed of multiple encoder blocks that function sequentially to refine the representation of the input sequence.

As the sequence progresses through the multi-head attention layer of the l-th encoder, the model generates a set of queries $Q$, keys $K$, and values $V$ for each attention head within the layer. These elements are essential components of the attention mechanism, serving to guide the model in identifying which parts of the sequence are relevant to each other. The computation of the attention outputs can be represented as follows:

$$Q_i^l = EW_i^q \tag{5}$$

$$K_i^l = EW_i^k \tag{6}$$

$$V_i^l = EW_i^v \tag{7}$$

attention output for each attention block,

$$Z_i^l = soft\max \frac{Q_i^{lK}{}_i^{l\,T}}{\sqrt{d_k}} V_i^l \tag{8}$$

where $d_k$ represents the number of columns of the $Q, K$ matrices, that is, the vector dimension. In a multi-head attention framework, each head computes its attention output independently, focusing on different parts of the sequence. The outputs of all heads are then concatenated to form a comprehensive representation, and finally the spliced outputs are passed through a linear layer to obtain the final representation $Z^l$:

$$Z^l = concat(Z_1^l, Z_2^l, \ldots, Z_n^l)W_z^l \tag{9}$$

where $W_z^l$ is the weight parameter generated by the linear layer, and n is the number of attention blocks. This output $Z^l$ is then added to the input $E^l$ in a residual-like manner, akin to the residual connections found in ResNet (*He et al., 2016*) architectures. The combined output is normalized through a layer normalization process, after which the resultant vector is passed through a fully connected layer with a *Relu* (Rectified Linear Unit) activation function. Finally, the process concludes with another residual connection and a subsequent layer normalization to ensure the preservation and enhancement of information throughout the encoding layers.

$$U^l = LayerNorm(E^l + Z^l) \tag{10}$$

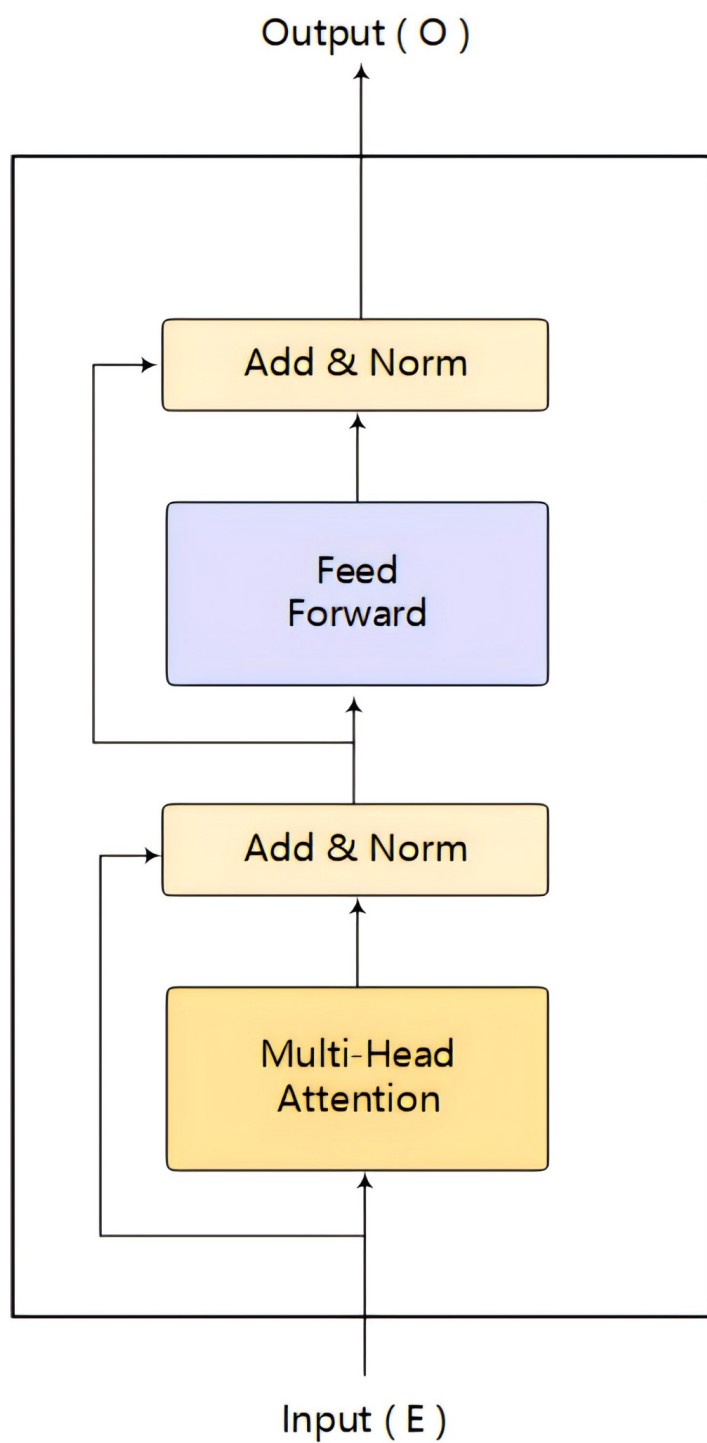

**Figure 3** Transformer Encoder single encoder model.

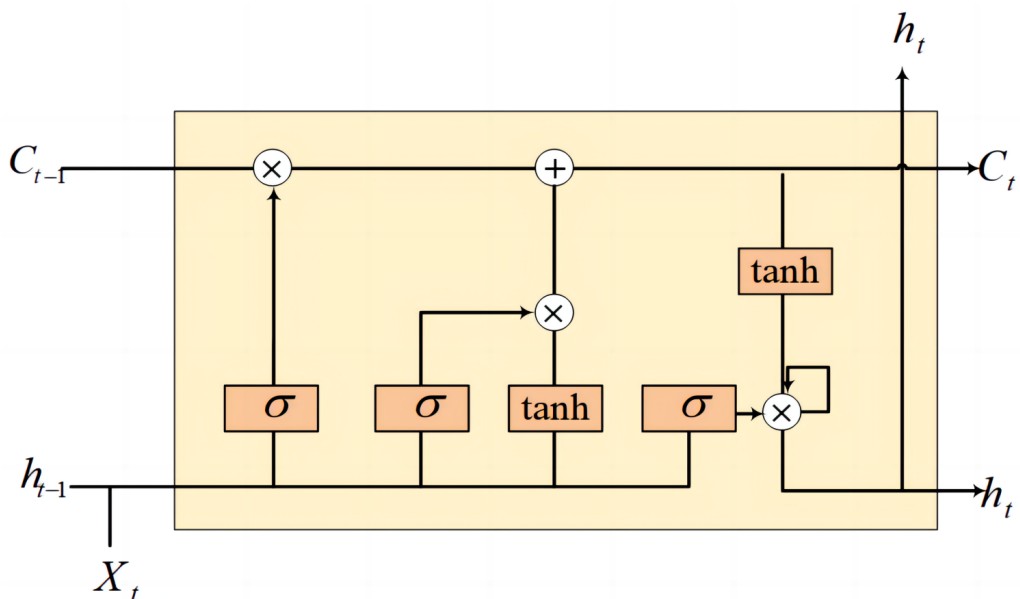

**Figure 4** LSTM cell structure.

$$O^l = LayerNorm(\text{ReluU}^l(W_U^l) + U^l) \tag{11}$$

where $W_U^l$ is the weight parameter of the fully connected layer, and then the output of the previous encoder is used as the input of the next encoder, that is $E^{l+1} = O^l$. The Transformer encoder adeptly captures salient information within opcode sequences, providing a foundation for subsequent network layers to extract features.

**Bi-GRU:** Recurrent neural network (RNN) (*Zaremba, Sutskever & Vinyals, 2014*) are inherently designed for sequence processing, adept at capturing temporal dependencies by fusing the output from a previous timestep with the current input to form a continuous loop of information flow. Traditional RNN architectures, however, face significant challenges when dealing with long sequences due to the propensity for gradients to either explode or vanish—an issue that impedes the learning of long-distance dependencies.

To surmount this challenge, long short-term memory (LSTM) (*Hochreiter & Schmidhuber, 1997*) networks were developed with a sophisticated gating mechanism that governs the retention and omission of information across long sequences. This mechanism enables LSTM to selectively maintain or discard information through a series of gates—namely, the input, forget, and output gates—complemented by a cell state that carries long-term information through the network. The LSTM unit structure is shown in the Fig. 4.

The gated recurrent unit (GRU) (*Chung et al., 2014*), on the other hand, is an iteration on the LSTM design, streamlining the architecture by combining the cell and hidden states, and reducing the gating system to two gates: the update gate and the reset gate. This simplification leads to fewer parameters, potentially easing the computational load and

expediting the training process. Despite this simplification, GRU maintain a competitive edge in capturing relevant features from long sequences, similar to LSTM.

Figure 5 can be represented by the formula as:

$$h_t = (1 - z_t) \odot h_{t-1} + z_t \odot \widetilde{h}_t. \tag{12}$$

In the realm of GRU, the hidden state at any given time, denoted as $h_t$, is a critical component that encapsulates the learned information up to that point in the sequence. Concurrently, $\widetilde{h}_t$ signifies the candidate hidden state, which represents a possible new value for $h_t$. In a bi-GRU, the sequence processing is further enhanced by incorporating two separate GRU: one processes the sequence from start to end, while the other processes it in reverse, from end to start. This bidirectional approach allows the network to capture dependencies from both future and past contexts, providing a more robust representation of the sequence.

$$z_t = \sigma(W_z \cdot [h_{t-1}, x_t]). \tag{13}$$

The update gate, represented by $z_t$, plays a pivotal role in mediating the flow of information between the past and the current state. It operates by determining the degree to which the previous hidden state, $h_{t-1}$, should be retained and how much the candidate hidden state, $\widetilde{h}_t$, should be incorporated into the current hidden state, $h_t$. The update gate effectively decides the balance of preserving historical information against admitting new insights. where the symbol ([]) represents a matrix concatenation operation. When the weight matrix $W_z$ is multiplied by the concatenation matrix formed by the previously hidden state $h_{t-1}$ and the current input $x_t$, it is transformed by the sigmoid function to obtain the value of the update gate $z_t$. The magnitude of $z_t$ governs the degree to which information from the past is preserved: a larger $z_t$ implies a higher retention of past information, whereas a smaller $z_t$ indicates a greater allowance for the assimilation of new information. This mechanism is pivotal for GRU in managing long-term dependencies, enabling them to adeptly capture information across extended sequences.

The candidate hidden state, which represents the new information that can be passed to the next time step in the context of GRU networks, can be expressed by the following equation:

$$\widetilde{h}_t = \tanh(W_h \cdot [r_t \odot h_{t-1}, x_t]). \tag{14}$$

Here, $W_h$ is the weight matrix associated with the candidate hidden state, represents the concatenation of the previous hidden state $h_{t-1}$ and the current input $x_t$, and *tanh* denotes the hyperbolic tangent activation function that helps to regulate the values of the candidate hidden state, ensuring they are within the range of $-1$ to $1$.

$$r_t = \sigma(W_r \cdot [h_{t-1}, x_t]). \tag{15}$$

Here, the value of $r_t$ influences the degree of retention or forgetting of past information: a higher $r_t$ signifies that more past information is conserved, whereas a lower $r_t$ leads to more forgetting. Further, the preserved information from the past, modulated by the reset gate, is then combined with current input *via* concatenation, and this composite

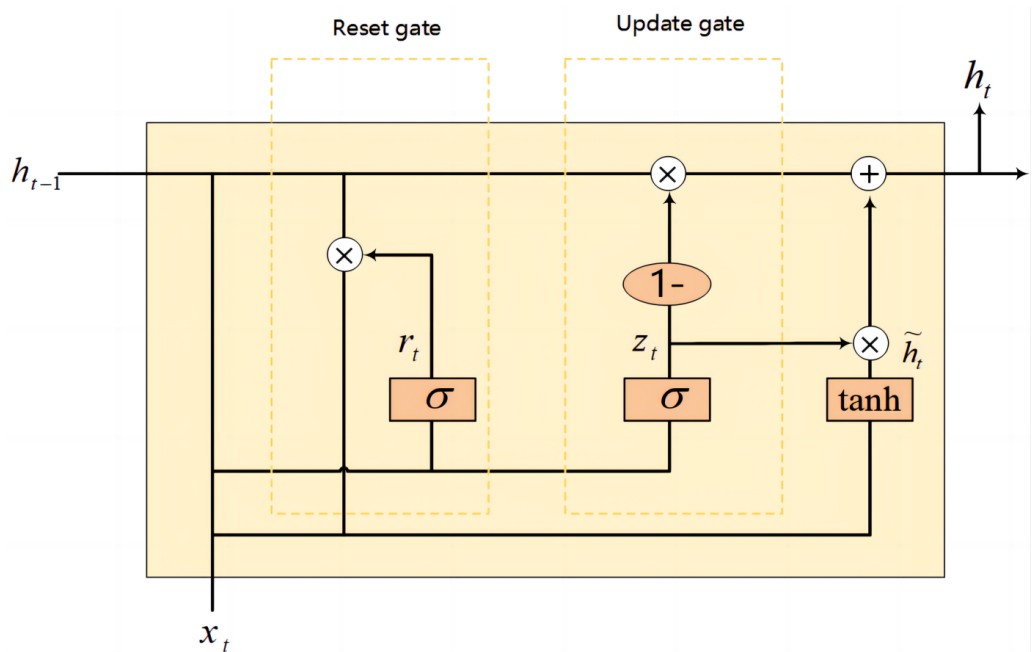

**Figure 5** GRU cell structure.

is subsequently multiplied by a weight matrix. Afterward, this product passes through the *tanh* activation function to produce the candidate hidden state $\widetilde{h}_t$. The candidate hidden state $\widetilde{h}_t$ serves as a provisional state, encapsulating a fusion of past information and current input. Subsequently, contingent upon the function of the update gate, it is determined whether to employ this candidate state to refresh the ultimate hidden state at the current moment. The gating mechanisms within the GRU not only enable the learning of information over long sequences but also address the issues of gradient vanishing and explosion that are common in traditional recurrent neural network (RNN) architectures.

However, the standard unidirectional GRU processes the input sequence in temporal order, with each hidden state depending on past information alone. In contrast, the bi-GRU takes into account both past and future information by introducing two separate processing directions: forward and backward. This bidirectional approach ensures that the hidden state at each time step encompasses the contextual information of the entire sequence, as shown in Fig. 6 of the bi-GRU structure. The bi-GRU is capable of discerning intricate patterns and regularities within opcode sequences, enabling the acquisition of the holistic structure and semantics of the opcode sequences.

The Transformer encoder, with its multi-head attention mechanism, demonstrates exceptional feature extraction capabilities at a deep level. Simultaneously, the bi-GRU offers comprehensive insights into opcode sequences through its bidirectional processing mechanism. This combined deep and broad strategy significantly enhances the model's efficiency and accuracy in extracting features from opcode sequences.

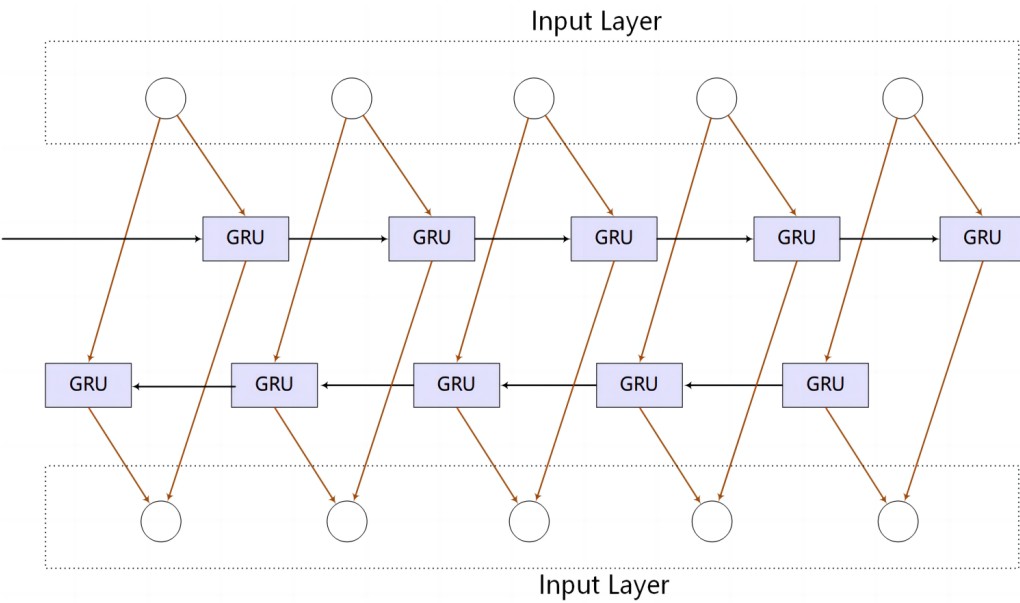

**Figure 6  Bi-GRU structure.**

# EXPERIMENT

## Dataset processing

Our study employs a dataset meticulously curated by *Rossini (2022)*, which was constructed starting from a list of verified smart contracts provided by Smart Contract Sanctuary, with reference to Smart Bugs (*Durieux et al., 2020*). Among these tools, Slither was used to detect four types of vulnerabilities in smart contracts, which include access-control, arithmetic, other, reentrancy, safe, unchecked-calls.

In this research, opcode is chosen as the focus for vulnerability analysis for several compelling reasons: (1) Source code is written by developers, leading to instances where different function names can perform identical operations, and the presence of extensive comments and whitespace may introduce significant noise, potentially impeding the model's capacity to extract meaningful information. (2) Smart contract bytecode tends to be unreadable, given its hexadecimal representation, making the extraction of sequential information using existing methods quite challenging. (3) There exists a one-to-one correspondence between opcodes and bytecode within smart contracts, effectively capturing the contract's logic. Therefore, leveraging opcodes is deemed more appropriate for this analysis. The relationship between source code, bytecode, and opcode is illustrated in the Fig. 7.

In our preprocessing phase, the source code within the dataset has been compiled into bytecode. This bytecode is subsequently mapped to smart contract opcodes following the guidelines provided by the Ethereum Yellow Paper (*Wood et al., 2014*), as shown in Table 1. The smart contract opcodes comprise 142 distinct operation instructions, categorized into 10 functionalities: arithmetic operations, block information operations, comparisons, stack

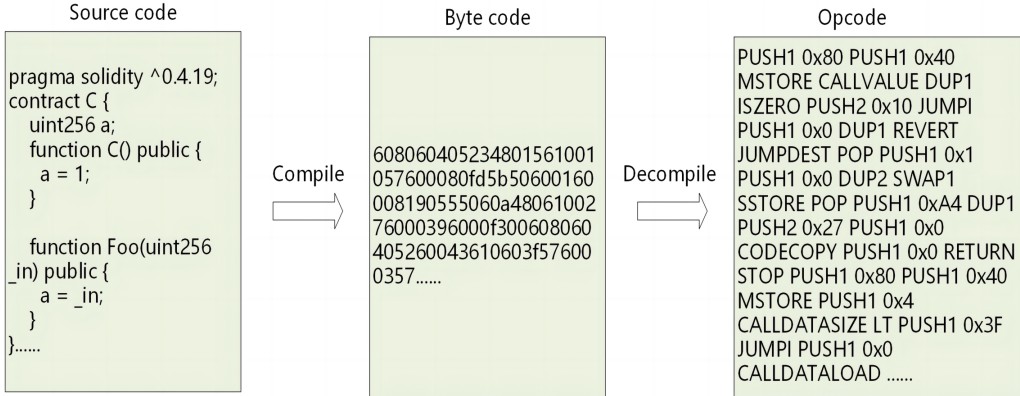

| Source code | | Byte code | | Opcode |

**Figure 7** The interrelationship between source code, bytecode, and opcode.

**Table 1** Partial bytecode to opcode correspondence.

| Bytecode | Opcode |
| --- | --- |
| 0×00 | STOP |
| 0×01 | ADD |
| 0×02 | MUL |
| 0×03 | SUB |
| 0×60 | PUSH1 |
| 0×80 | DUP1 |
| 0×90 | SWAP1 |
| 0×a0 | LOG0 |

instructions, memory access, storage handling, and jump instructions, among others. To circumvent the potential decline in model performance due to the extensive variety of instructions, which could hamper the model's ability to discern sequential information effectively, we adopted a simplification methodology inspired by *Huang et al. (2022)*. This approach involves streamlining the opcodes to facilitate more efficient model processing and improved interpretability. The simplified opcodes reduce the complexity of the opcode sequences, alleviate the model's burden, and allow the model to focus on a broader scope rather than specific instruction sequences. This helps the model perform better on unseen data, enhancing its generalizability. The simplification process and its outcome are detailed in Table 2.

To ascertain the efficacy of our model, we conducted comparative analyses against other deep learning-based vulnerability detection models within this research, incorporating ablation experiments as well. Each dataset utilized in the experiments was partitioned in a 9:1 ratio to create distinct sets for training and validation.

## Experimental environment and experimental parameter settings

The experimental environment settings are delineated in Tables 3 and 4. In the parameterization of the experiments, we employ Dropout as a regularization technique

**Table 2  Simplified opcode correspondence table.**

| Simplified Opcode | Original Opcode |
| --- | --- |
| PUSH | PUSH1-PUSH32 |
| DUP | DUP1-DUP16 |
| SWAP | SWAP1-SWAP16 |
| LOG | LOG0-LOG4 |

**Table 3  Experiment environment settings.**

| Software and Hardware | Allocation |
| --- | --- |
| Operating system | Ubuntu |
| GPU | NVIDIA GeForece RTX4070ti |
| Memory | 32GB |
| Disk | 1T |
| CUDA | 12.2 |
| Pytorch | 2.1.1+cu121 |
| Python | 3.10.10 |

**Table 4  Experimental parameter settings.**

| Parameter | Setting |
| --- | --- |
| Learning rate | 1e−5 |
| Dropout rate | 0.5 |
| Optimizer | Adam |
| Loss function | Asymmetric Loss |
| Epoch | 20 |

to mitigate the risk of overfitting and enhance the generalization capability of the model. During each training iteration, Dropout randomly selects a subset of neurons, setting their outputs to zero, which prompts the remaining neurons to partake in forward and backward propagation. This method helps prevent the network from becoming overly reliant on any particular set of neurons, thereby reducing the potential for overfitting.

There are various optimizers available, such as the commonly used Stochastic Gradient Descent (SGD) and Adaptive Moment Estimation (Adam). Our research utilizes the Adam optimizer, principally due to its incorporation of both momentum and adaptive learning rate properties. Adam is capable of estimating the first and second moments of the gradient, applying these estimates to the parameter update process, which can often lead to more efficient convergence. For loss function selection, we adopt the asymmetric loss proposed by *Ridnik et al. (2021)* to address data imbalance in multi-label classification.

## Evaluation metrics

Given that the present research addresses a multi-label classification problem, the conventional accuracy metric is not an optimal measure for evaluating multi-label classifications. Therefore, this research employs $HammingLoss$, $micro-P$, $micro-R$,

and the $micro - F1$ to assess the performance of the multi-label vulnerability detection model. The calculations for each metric are as follows:

$$micro - R = \frac{\sum_{i=1}^{n} TP_i}{\sum_{i=1}^{n} TP_i + \sum_{i=1}^{n} FN_i} \qquad (16)$$

$$micro - P = \frac{\sum_{i=1}^{n} TP_i}{\sum_{i=1}^{n} TP_i + \sum_{i=1}^{n} FP_i} \qquad (17)$$

$$micro - F1 = \frac{2 \times (micro - R) \times (micro - P)}{(micro - R) + (micro - P)} \qquad (18)$$

where $n$ represents the total number of label predictions made, which is equal to the sample count multiplied by the number of label categories. $TP$ (true positive) refers to a correctly predicted positive instance where the model has accurately identified the presence of a label. $FN$ (false negative) is when the model incorrectly predicts the actual absence of a label. $FP$ (false positive) is a situation where the model incorrectly predicts the presence of a label that doesn't actually exist. $TN$ (true negative) refers to the correctly predicted negative instances, where the model has accurately identified the absence of a label.

The *HammingLoss* is a performance metric that is particularly well-suited to the evaluation of multi-label classification problems. In such problems, each sample can be associated with multiple labels rather than a single label. The *HammingLoss* focuses on assessing the level of disagreement between the predicted label set and the true label set for each sample. *HammingLoss* is calculated by determining the proportion of incorrectly predicted labels over all labels and is represented as the average fraction of labels that are not correctly predicted. If a model's predicted label set for a sample is nearly identical to the true label set, the *HammingLoss* will be low, indicating good model performance. Conversely, if there is a large disparity between the predicted and true labels, the *HammingLoss* will be high, indicating poor model performance. The formula for *HammingLoss* is given by:

$$HammingLoss = \frac{1}{N \times L} \sum_{i=1}^{N} \sum_{j=1}^{L} [y_{ij} \oplus \widehat{y}_{ij}]. \qquad (19)$$

Here, $N$ is the number of samples, $L$ is the number of labels, $y$ represents the true value, and $\widehat{y}$ represents the predicted value. The result is 1 if and only if the predicted value and the true value are consistent; otherwise, the result is 0. As shown in Table 5.

## Experimental results

**Experiment 1:** In this experiment, we aim to verify the effectiveness of the proposed model. To this end, we compared the proposed Ext-ttg model with several popular deep learning models, including LSTM, Bi-LSTM, LSTM-ATT, and TextCNN. These models were evaluated on the same dataset in the comparative experiment.

The comparison results, as shown in Table 6, clearly demonstrate the superior performance of the Ext-ttg model across multiple performance evaluation metrics.

**Table 5  Confusion matrix.**

| Predicted value | Actual value | |
| --- | --- | --- |
| | **0** | **1** |
| 0 | TN | FP |
| 1 | FN | TP |

**Table 6  Model evaluation results.**

| Model | Micro-R(%) | Micro-P(%) | Micro-F1(%) | *HammingLoss* |
| --- | --- | --- | --- | --- |
| LSTM | 73.81 | 82.53 | 77.92 | 0.125 |
| Bi-LSTM | 74.60 | 84.11 | 79.11 | 0.118 |
| Bi-LSTM-ATT | 74.98 | 85.32 | 79.81 | 0.114 |
| TextCNN | 72.13 | 86.35 | 78.60 | 0.119 |
| Ext-ttg | 79.61 | 85.57 | 82.48 | 0.107 |

Specifically, the Ext-ttg model achieved a micro-recall (Micro-R) of 79.61%, a micro-precision (Micro-P) of 85.57%, and a micro-F1 score (Micro-F1) of 82.48%, which are significantly better than the other models compared. These results indicate that our model can more balancedly recognize and classify samples in the experimental dataset while considering both recall and precision. In addition, the Ext-ttg model also excelled in the *HammingLoss* metric, recording a low rate of 0.107, which is far below the loss values of other models, suggesting that the model has a lower frequency of misclassification during the labeling process. TextCNN outperforms other models in terms of the MAR-P (micro-averaged recall at precision) metric, but it has the lowest recall rate. Upon analysis, it was found that the TextCNN model tends to produce more false positives, which explains the relatively low recall rate of the TextCNN model.

Overall, the Ext-ttg model outperforms the other models in the control group across all evaluation metrics. These findings validate our hypothesis regarding the effectiveness of the Ext-ttg model in processing this dataset. Its exceptional performance can be attributed to the design of the model architecture, which likely incorporates more effective mechanisms for capturing and processing information, such as an attention mechanism or specific architectural optimizations that enhance the recognition of textual features. These results provide a valuable reference for subsequent research and indicate that the Ext-ttg model has broad application prospects in similar tasks.

**Experiment 2:** To analyze the role of extractive summarization within our Ext-ttg model, we conducted a second experiment—an ablation research. In this ablation research, we designed one variant: the ttg (crop 2048) model, which trims the opcode sequence to a length of 2048, padding the sequence where necessary. The method removed the step of extractive summarization to observe its impact on model performance. According to the experimental results, as shown in Table 7, we observed the following:

In the absence of extractive summarization, the ttg (crop 2048) model's corresponding performances were 77.22%, 84.75%, and 80.81%. Compared to the models without

| Table 7 | Ablation research results. | | | |
|---|---|---|---|---|
| Model | Micro-R(%) | Micro-P(%) | Micro-F1(%) | *HammingLoss* |
| ttg (crop 2048) | 77.22 | 84.75 | 80.81 | 0.110 |
| Ext-ttg | 79.61 | 85.57 | 82.48 | 0.107 |

extractive summarization, our Ext-ttg model showed a significant improvement in these metrics, achieving 79.61%, 85.57%, and 82.48%.

Regarding the *HammingLoss* metric, models without extractive summarization exhibited a higher loss value of 0.110. In contrast, the Ext-ttg model performed significantly better, at only 0.107, indicating a higher frequency of classification errors when extractive summarization was removed from the process.

The results underscore the crucial role of extractive summarization in the Ext-ttg model, notably enhancing performance across metrics and reducing classification errors. Extractive summarization refines input data, aiding focused learning and key feature extraction, thus boosting model efficiency. In summary, the ablation research validates our hypothesis on the significance of extractive summarization in the Ext-ttg model's superior performance.

## CONCLUSION AND FUTURE WORK

Multi-vulnerability identification in smart contracts has long been a challenging issue, with previous models typically focusing on detecting single vulnerabilities. In our work, we propose a smart contract vulnerability detection model that combines extractive summarization methods with deep learning to identify multiple vulnerabilities in smart contracts. By employing extractive summarization methods to truncate opcode sequences, we reduce their length while preserving crucial sequences. Subsequently, we use a stacked transformer encoder and Bi-GRU for block-level and global feature extraction. Compared to existing methods, our approach captures more opcode information while retaining important opcode details, ensuring good model detection accuracy. This indicates the effectiveness and superiority of our proposed model in handling multi-label vulnerability detection tasks in smart contracts.

While this research has achieved certain results in the detection of vulnerabilities in smart contracts, there are still challenges that must be faced. Identifying the complex relationship between smart contract code and potential vulnerabilities remains a core difficulty in the field and warrants further exploration. In the future, we plan to focus on the following areas to further optimize our method: (1) We will attempt to incorporate expert rules to enable the model to extract richer semantic information. (2) We will explore more efficient model structures to detect vulnerabilities in smart contracts. We are confident that the methods proposed by us will provide valuable insights and references for research in the field of smart contract vulnerability detection.

## ACKNOWLEDGEMENTS

We express our heartfelt appreciation to the reviewers for their constructive comments during the manuscript review process.

### Funding

This work has been supported by the General Project of Science and Technology Program of Jiangxi Provincial Department of Education (NO. GJJ200721); the Jiangxi Provincial Radioactive Geoscience and Big Data Technology Engineering Laboratory Open Fund (JELRGBDT201709, JELRGBDT202207), the Jiangxi Provincial Key Laboratory of Cyberspace Security Intelligent Perception Open Fund Project (JKLCIP202211), and the Jiangxi Provincial Graduate Innovation Special Fund Project (DHYC-202336). The funders provided assistance in research design, data collection and analysis, publication decisions, and manuscript preparation.

### Grant Disclosures

The following grant information was disclosed by the authors:
The General Project of Science and Technology Program of Jiangxi Provincial Department of Education: NO. GJJ200721.
Jiangxi Provincial Radioactive Geoscience and Big Data Technology Engineering Laboratory Open Fund: JELRGBDT201709, JELRGBDT202207.
Jiangxi Provincial Key Laboratory of Cyberspace Security Intelligent Perception Open Fund Project: JKLCIP202211.
Jiangxi Provincial Graduate Innovation Special Fund Project: DHYC-202336.

### Competing Interests

The authors declare there are no competing interests.

### Author Contributions

- Zhibo Wang conceived and designed the experiments, performed the experiments, analyzed the data, performed the computation work, prepared figures and/or tables, and approved the final draft.
- Liu Guoming conceived and designed the experiments, performed the experiments, analyzed the data, performed the computation work, prepared figures and/or tables, and approved the final draft.
- Hongzhen Xu conceived and designed the experiments, authored or reviewed drafts of the article, and approved the final draft.
- Shengyu You conceived and designed the experiments, authored or reviewed drafts of the article, and approved the final draft.
- Han Ma analyzed the data, authored or reviewed drafts of the article, and approved the final draft.
- Hongling Wang analyzed the data, authored or reviewed drafts of the article, and approved the final draft.

### Data Availability

The code is available at GitHub and Zenodo:

- https://github.com/LGM-233/model.

Guoming L. 2024. Deep Learning-Based Methodology for Vulnerability Detection in Smart Contracts [Data set]. Zenodo. https://doi.org/10.5281/zenodo.13418255

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
