# Peer review of "Deep learning-based methodology for vulnerability detection in smart contracts"

_PeerJ Computer Science, doi:10.7717/peerj-cs.2320_

## Round 0.1 · original submission · Major Revisions

Dear Authors, upon reviewing the paper we believe it is of potential interest to the readership of PeerJ Computer Science and tackles a timely problem, but it requires significant revisions to enhance clarity, comprehensiveness, and reproducibility. In particular, please address methodological concerns raised by the reviewers including (i) clearly defining the research question and the specific gap addressed, (ii) providing detailed descriptions of the process and data processing methods to ensure replicability, (iii) testing the method on multiple datasets, compare it with other specific techniques in the literature and highlighting specific threats to validity of the study.

**Language Note:** The review process has identified that the English language must be improved. PeerJ can provide language editing services - please contact us at [email protected] for pricing (be sure to provide your manuscript number and title). Alternatively, you should make your own arrangements to improve the language quality and provide details in your response letter. – PeerJ Staff

Reviewer 1 ·

Basic reporting

no comment

Experimental design

no comment

Validity of the findings

Thank you for providing supplemental data and python code. However, the supplemental files are missing instructions alongside documentation of dependent libraries on how to run and reproduce the experimental results from the paper. These should be included to help future readers validate the findings in the paper.

Additional comments

Minor Comments:
Line 149: B_num is not properly subscripted.
Line 160: Figure 1 needs more information as to what exactly are the relationships between the example opcode sequences presented as B1,B2,B3,B4. As of now, it merely shows the pairwise relationship of the nodes. Consider including the computed similarity value onto the edges of figure 1.
Equation 9: Shouldn’t LHS be Z^l and not Z^l_i?

Cite this review as

Reviewer 2 ·

Basic reporting

This paper describes a new process for classifying vulnerabilities in smart contracts based on analyzing opcode sequences. The process was built using a sequence of analysis operations based on sequence extraction and a custom neural network-based process (Transformer+Bi-GRU). The system was trained and evaluated using an annotated dataset from the literature. The results demonstrate better performance compared to other neural network-based systems evaluated.

The overall English proficiency is fairly adequate, though it varies considerably between sections. The "Enhanced Ocode Pruning" section warrants re-examination. For example, on page 3 line 148, the sentence "Bv is constructed by to a vector dimension of 100" contains some grammatical issues.
The punctuation throughout should be double-checked (sometimes a comma is used instead of a period), as well as footnotes and subscript.
The "Experiments" section paragraph titled "Dataset Processing" could benefit from revision for clarity. For example, in page 9, on line 263 it states the source code is first decompiled to bytecode, but it should be the reverse. It then says the opcodes are generated according to the guidelines of the yellow paper, but this operation can be performed using appropriate tools.

The introduction outlines the motivations and context. However, it could be strengthened from the perspective of discussing how the work is situated within the problem space. Providing how this research builds upon or differs from prior studies could help contextualize the contribution and significance of the presented findings.
When reviewing related works, more recent findings regarding neural network-based approaches should be cited (for example: 10.1038/s41598-023-47219-0). The novel aspects introduced in this paper compared to the existing literature are not fully clarified. Including the most up-to-date references will help ensure a comprehensive overview of progress in this area.

The overall structure of the paper is reasonable.
It may help to re-evaluate whether Figure 1 is necessary and, if so, provide more context for interpretation. Figure 2 could benefit from reviewing and enhancing the label, description, and alignment between the graphic and textual explanation to better facilitate comprehension.

Experimental design

This research appears novel and well-aligned with the aims and scope of the journal.

While the research question is not explicitly defined, its aim of improving the performance of smart contract vulnerability identification can be inferred. However, the specific gap in knowledge or understanding that this work seeks to address is not expressly stated.

The authors aim to discuss the methodology in depth. However, it could be improved in terms of clarity and completeness. In particular, despite the numerous technical details regarding the method components, the motivations behind the choices and practical details of the process are not fully explained.
For example, it is described that the process is based on analyzing "sequences" of opcodes, but it is not clear how these sequences are defined and why they can be up to 8,000 characters long. Additionally, Skip-gram is established as the method but the rationale for this choice is not explained. The pruning part is a bit confusing and it is unclear what Bv and the Bvi represent. The part regarding the neural network model is discussed better but the execution flow of the process remains unclear.
In the Experiments section, an additional element is added to the process: simplifying the opcodes. It would be helpful to understand if this procedure actually improves the results.

Given the lack of clarity around the process and data processing methods, the experiment is not directly replicable. More details on key methodological choices and their implementation would strengthen understanding and evaluation.

Validity of the findings

While the experiments are reasonably conducted by comparing the proposed method to established baselines using a single dataset, some extensions could strengthen the evaluation. It would be advisable to test the method on multiple datasets and compare against other literature-specific techniques for detecting vulnerabilities in Solidity smart contracts. The results could also highlight whether classification performance is consistent across all vulnerability types or not. Ultimately, more context is needed to fully assess the real-world efficacy of the method for this specific application. While a good starting point, more extensive experimentation could provide stronger evidence regarding the method's effectiveness and limitations within this technical domain.

Regarding the data used, it is worth noting that in the reference dataset, the labels were generated using a separate static analysis tool. It would therefore be useful to leverage data from multiple datasets, as done in other related works.

The conclusions could better highlight the specific contributions of the paper, as they are currently focused primarily on describing the context and methodology. The section on open problems would benefit from rewriting to more directly connect it back to the work presented Drawing explicit links between the research questions, methodology, results and their novel implications would help strengthen the concluding remarks.

Cite this review as

---

## Round 0.2 · accepted · Accept

We appreciate your efforts in addressing the feedback provided by the reviewers. As noted in the referee report, your revisions have improved the clarity and consistency of the paper. The referee has highlighted the importance of sharing code for reproducibility (also in light of the revisions to results and algorithm after the first round of revision). To this end, we kindly request that you include a reference to the source code used in your experiments. Sharing your code will greatly enhance the reproducibility of your findings and contribute to the broader scientific community.

Please add a link to the code repository in your final version of your manuscript. Please also include a working link to the dataset used (in ref [23]) .

Reviewer 2 ·

Basic reporting

The authors have welcomed the suggestions to improve the paper, which has now gained clarity in all its sections. There are still problems with formatting (check spaces after periods).

Experimental design

The design of the experiments is now more consistent, and the process is clearer. Inaccuracies in the explanation of the process have been corrected. The review work revealed an error in the process that invalidated the obtained results. Due to the poor performance of the corrected results, the authors intervened in the process and successfully optimized the outcomes. In particular, opcode simplification is now reserved for simple cases only.

Validity of the findings

In their answers, the authors state that their work could serve as a reference for future research. For this reason, they should include a reference to the source code to ensure complete reproducibility of their work.

Cite this review as